# How Much Stress Does a Surgeon Endure? The Effects of the Robotic Approach on the Autonomic Nervous System of a Surgeon in the Modern Era of Thoracic Surgery

**DOI:** 10.3390/cancers15041207

**Published:** 2023-02-14

**Authors:** Antonio Mazzella, Monica Casiraghi, Domenico Galetta, Andrea Cara, Patrick Maisonneuve, Francesco Petrella, Giorgio Lo Iacono, Eleonora Brivio, Paolo Guiddi, Gabriella Pravettoni, Lorenzo Spaggiari

**Affiliations:** 1Division of Thoracic Surgery, IEO, European Institute of Oncology, IRCCS, 20141 Milan, Italy; 2Division of Epidemiology and Biostatistics, IEO, European Institute of Oncology, IRCCS, 20141 Milan, Italy; 3Applied Research Division for Cognitive and Psychological Sciences, IEO, European Institute of Oncology, IRCCS, 20141 Milan, Italy; 4Department of Oncology and Hemato-Oncology, University of Milan, 20141 Milan, Italy

**Keywords:** robotic surgery, lung cancer, thoracic surgery, cardiovascular system, surgical stress, autonomic nervous system

## Abstract

**Simple Summary:**

A surgeon’s feelings and his/her autonomic nervous system (ANS) response during interventions represent direct indicators of comfort, comfortability, dexterity, and stress during a surgical procedure. We evaluated the autonomic nervous system (ANS) and psychological responses to stress of surgeons during their surgical activity, comparing their robotic activity and their classical surgical activity via an open approach. For different reasons, the robotic approach led to less stimulation of the autonomic nervous system, producing less stress for the surgeons and ensuring greater comfort.

**Abstract:**

(1) Objective: the purpose of this study was to evaluate and quantify the stress to which a surgeon is subjected during his/her surgical activity; we compared the individual clinical and psychological responses to stress of two surgeons during their surgical activities via robotic and open approaches. (2) Materials and methods: This was a prospective observational study in which we progressively collected data concerning the surgical performances of two different thoracic surgeons (October 2021–June 2022). We evaluated 20 lung resections performed via robot-assisted surgery and 20 lung resections performed via an open approach by each surgeon; in particular, we evaluated a panel of pre-, peri-, and postoperative data concerning the interventions, the patients, and other outcomes concerning the autonomic nervous system (ANS) and psychological responses to stress of the surgeons during their surgical activities. (3) Results: When analyzing data concerning the ANS activity of two surgeons, during robotic activity we found lower maximum, minimum, and mean heart rates; lower mean respiratory frequencies; lower body temperatures; and lower mean desaturations compared to the open approach activity for both surgeons. The psychological monitoring showed that the open approach created more physical fatigue and frustration but higher levels of satisfaction and performance evaluation. The robot-assisted surgeries showed higher levels of anxiety. (4) Conclusions: for different reasons, the robotic approach stimulated the ANS to a lesser degree, causing less stress for surgeons and ensuring greater comfort.

## 1. Introduction

Minimally invasive and robot-assisted thoracoscopic surgery (RATS) currently represents a cornerstone in the surgical treatment of early-stage non-small-cell lung cancer (NSCLC). From the first report on a robot-assisted lobectomy performed by the da Vinci surgical system in 2002 [1], we have witnessed increased confidence in the technique and its technical improvement thanks to the advent of more manageable machines (da Vinci XI) and new devices (robotic staplers and energy devices). 

The advantages of minimally invasive and robotic surgery have been widely discussed in the literature, and at present, robotic lobectomy is recognized as being a feasible and safe approach [2,3,4,5], with clinical outcomes similar, or in some cases superior, to those of video-assisted thoracoscopic surgery (VATS) [6,7,8,9], even in advanced stages (II-IIIA NSCLC) [10].

Nevertheless, in some cases open surgery represents the only sure and safe therapeutic approach with guaranteed oncologic results; the open approach is currently reserved for more complex cases characterized by vascular reconstruction, chest wall involvement, or the adjacency or infiltration of mediastinal structures.

The surgeon’s feelings and his/her autonomic nervous system (ANS) response during the interventions represent direct indicators of comfort, comfortability, dexterity, and stress during a surgical procedure. However, the effect of this stress for the surgeon has been infrequently and incompletely evaluated in the literature.

Surgeons, in comparison to medical physicians, consider their work to be physically strenuous and made harder by uncomfortable and exhausting positions and stressful situations [11,12,13]. The physiological impact of surgical stress can be measured via heart rate and heart rate variability (HRV), which are considered to be consequences of ANS activity, regulating the time interval (milliseconds) between heartbeats [14,15]. Low HRV and a higher heart rate (HR) are associated with more stressful events and more stressful interventions [15,16]. A recent paper related blood pressure, heart rate, O2 saturation, and CO_2_ production with the activation of the ANS during surgery [17]. 

In this era in which thoracic surgery and oncology present, day after day, new therapeutic innovations (biological and immunotherapy) with new technical difficulties for surgeons, it seems very interesting to evaluate the cardiovascular and psychological stresses during an intervention. It is even more interesting to understand how different surgical approaches can determine very different responses in the same surgeon.

The purpose of this study was to evaluate and quantify the emotional arousal stress and general comfort to which a surgeon was subjected during his/her surgical activity. In addition, we aimed to compare the clinical and psychological responses to stress of a surgeon during his/her surgical activity via robotic and open approaches.

## 2. Materials and Methods

This was a prospective observational study that started in October 2021 and ended in June 2022. No authorization from our ethics committee was required because the device that was used was already approved in real life for the evaluation of vital parameters, both in basal conditions and under stress. 

This study was conducted in accordance with the ethical principles of the Declaration of Helsinki. We progressively collected data concerning the surgical performances of two different male thoracic surgeons in the period from October 2021 to June 2022. We evaluated 20 progressive lung resections via the robot-assisted approach and open surgery performed by two surgeons. In particular, we evaluated a panel of pre-, peri-, and postoperative data concerning the interventions and the cardiovascular activity and stress of the surgeons during their surgical activities.

### 2.1. Patient Work-Ups

For all patients, pretreatment work-ups included a total-body contrast-enhanced CT scan (brain, thorax, and abdomen) and a positron emission tomography scan. The mediastinal nodes were considered negative at clinical staging if the short axis was less than 1 cm and there was no significant (standardized uptake value <2.5) 18fluoro-2-deoxy-2-D-glucose uptake. Global spirometry (with DLCO/VA calculation) and a complete cardiological assessment were routinely performed. A preoperative histological diagnosis was obtained, if possible, via biopsy (fiberoptic bronchoscopy or fine-needle ago-biopsy (FNAB)). All patients were then discussed at our multidisciplinary meeting before the operation.

### 2.2. Selection Criteria for Robotic or Open Approaches

In our experience, the robotic approach was reserved for:-All I stage (IA1, IA2, IA3, and IB) cases without clinical lymph node involvement (cN0);-All clinical N1 stage (T1a, b, c to T2a, b, cN1)–IIB stage cases.

We performed lung resections with a traditional open approach in cases of:-Neoadjuvant therapy (chemo- or immunotherapy for IIIA-B preoperative staging);-Tumor dimensions > 5 cm, chest wall involvement, or adjacency to mediastinal structures (IIB and IIIA).

### 2.3. Surgical Technique (RATS and Open Approach)

Our technique and robotic approach have been described previously [18]. All procedures were performed under general anesthesia with a double-lumen endotracheal tube. Patients and personnel were positioned as previously reported [19]. With the patient in lateral decubitus, a 3 cm utility incision was performed at the fourth or fifth intercostal space, anteriorly. Through this incision, an 8 mm 30° tridimensional robotic endoscope was inserted into the chest to explore the pleura and help incise the other three 8 mm ports under direct vision: a camera port in the seventh or eighth intercostal space on the midaxillary line and two other 8 mm ports at the seventh or eighth intercostal space in the posterior axillary line and at the auscultatory triangle. We used the da Vinci XI system, and the robot was usually driven over the patient’s shoulder at a 15° angle and attached to the four ports. The surgical cart was docked from the left side of the patient for right or left thoracic procedures. The camera could be moved between two different ports, allowing a better view. We used the EndoWrist stapler (30 vascular and 30 or 45 parenchyma), which could be placed through a 12 mm port (for an inferior lobectomy or a segmentectomy, it was placed either at the utility thoracotomy or the posterior axillary line; for an upper lobectomy, it was placed at the posterior axillary line), as previously reported [20]. The fissure-last approach was always used. Portal placement did not change with the type of resection or the side. 

On the other hand, the open approach consisted of a classic muscle-sparing lateral thoracotomy (the latissimus dorsi was not cut but was spread) at the 5th intercostal space with rib spreading. At the end of surgery, a 28 CH soft chest tube was placed, and the thoracic cavum was closed with two pericostal sutures.

### 2.4. Patient Outcomes

We evaluated various pre- and postoperative parameters (Table 1). In particular, we considered the clinical characteristics and medical histories of the patients, the operative time (time from skin incision to the end of the surgical procedure), the types of interventions, conversion (unplanned extension of the incision and rib spreading), postoperative complications (according to the Clavien–Dindo classification), the postoperative hospital stay (from the day of the surgical procedure to the day of discharge), and the chest tube drainage duration.

### 2.5. Cardiovascular and Respiratory Activities of the Surgeons

The surgeons underwent full cardiological check-ups before starting the study. No cardiac/respiratory or general comorbidities were detected. No surgeon was taking medication at the time of the study.

The surgeons’ cardiac and respiratory parameters were measured using a wearable device: the Healer R2 (MDD Class IIa Medical Device developed by L.I.F.E. Italia S.r.l., Milan, Italy) (www.x10x.com; info@x10x.com, accessed on 1 November 2022), which is intended for multi-parametric monitoring applications (clinical or wellbeing) and diagnostic exams (polysomnography and ECG Holter) in hospital/outpatient and home settings. 

The device included a sensorized garment (Healer R2); a data logger for data acquisition, storage, and transmission with Wi-Fi or 4G connectivity; a cloud platform for data storage and elaboration; desktop software (Healer Desktop); a web portal (Healer Cloud); and two apps (Healer R and My Healer). 

The sensorized Healer R2 garment could record several physiological parameters and signals, each with its own sampling frequency (Figure 1 and Figure 2):

-Cardiovascular activity (mean, maximum, and minimum heart rates), thanks to a 6-lead ECG signal at 500 Hz using 4 ink-based dry electrodes;-Respiratory activity (mean, maximum, and minimum respiratory rates), desaturation, and time of desaturation, thanks to a 3-channel respiratory signal at 50 Hz from circumferential strain sensors placed at the thoracic, xiphoid, and abdominal levels;-Body activity and temperature (mean, maximum, and minimum body temperatures), thanks to a contact sensor under the right armpit;-Blood oxygen saturation (SpO2) (mean, maximum, and minimum SpO2 values, desaturation, and time in desaturation) from an optical module under the left armpit;-Activity level and body position from an inertial measurement unit (IMU) on the back.

We did not consider blood pressure as a parameter for different reasons. First of all, the mounting and pumping by a Holter device every 10 min could in itself cause stress to the surgeon. The second aspect is possible damage to an artery linked to the pumping. Lastly, continuous and intermittent pumping could compromise the surgical act, especially if prolonged and complex.

The surgeon wore the device for ten minutes half an hour before the start of surgery in order to monitor baseline parameters. He/she then wore the device for the entire intervention, from skin incision to the end of the surgical procedure.

### 2.6. Psychological Monitoring of the Surgeons’ State

The surgeons participating in the study completed the following questionnaires before and after each surgery:-STAI-Y1 is one of the scales of the STAI-Y (State-Trait Anxiety Inventory) [21,22], which aims to measure the presence and grade of anxiety. In particular, the STAI-Y1 rates the situation-related (that is, state) anxiety in the person filling the questionnaire. The questionnaire consists of 20 items with responses related to terms of intensity (from “almost never” to “almost always”) on a 5-point scale.-The Self-Assessment Manikin (SAM) is a non-verbal pictorial 5-point scale assessment technique that directly measures the valence (negative–positive) and arousal associated with a person’s affective reaction to a stimulus [23], providing a synthetic rating for emotional activation.-Perceived body part discomfort was measured on a 0 (no discomfort) to 10 (complete discomfort) scale. The considered body parts were the neck, shoulders (right and left separately), upper and lower back (separately), and right and left wrist/hand, as seen in similar studies [24].

Additionally, after each surgery, participants completed the NASA Task Load Inventory [25], a survey for quantifying workload on six different dimensions (mental, physical, and temporal demand; performance; effort; and frustration) on a 21-gradation scale.

### 2.7. Statistical Methods 

We compared the patient characteristics and surgeon levels of stress (heart rate, respiratory frequency, body temperature, and saturation) using Fisher’s exact test for categorical variables and Student’s *t*-test for continuous variables. The analyses were performed using SAS software version 9.4 (Cary, NC, USA). All *p*-values were two-sided. For psychological questionnaires, delta values between the pre- and post-operative interventions were calculated, taking into account the differences in the kinds of operatory approach (open vs. robotic). Only descriptive statistics were used because of the small sample size of the study. 

## 3. Results

Surgeon A (AM) and surgeon B (DG) performed 20 open approach and 20 robotic anatomical lung resections in the observation period. 

The clinical characteristics of the patients, types of interventions, complications, and postoperative outcomes are reported in Table 1.

Only one conversion in thoracotomy was observed because of technical anesthesiologic difficulties (inadequate pulmonary exclusion). No 30- or 90-day mortality was observed in the two groups. 

The surgeons’ cardiovascular and respiratory outcomes are described in Table 2.

Neoadjuvant chemotherapy was more often administered prior to open surgery than RATS (*p* < 0.0001). A higher proportion of patients in the open surgery group had lung comorbidities than in the RATS group (55% vs. 30%, respectively, *p* = 0.04). Patients who received RATS had significantly shorter durations of drainage (*p* = 0.003) and hospitalization (*p* < 0.0001) than patients who underwent open surgery.

The duration of intervention was significantly shorter in the RATS group than in the open surgery group (*p* = 0.02), as patients in the open group had more complex/extended surgeries.

When analyzing the data concerning the ANS activity of the two surgeons, during robotic activity we found (for both surgeons) lower maximum (*p* > 0.001 for surgeon A, *p*: 0.003 for surgeon B), minimum (*p* > 0.001 for both surgeons), and mean (*p* > 0.001 for surgeon A, *p*: 0.04 for surgeon B) heart rates; lower mean respiratory (*p* > 0.001 for surgeon A, *p*: 0.002 for surgeon B) frequencies; lower maximum body temperature (*p* > 0.001 for surgeon A, *p*: 0.008 for surgeon B); and lower mean desaturation (*p*: 0.08 for surgeon A, *p*: 0.04 for surgeon B) compared to open approach activity (Table 2). 

Both the STAI-Y and the SAM approaches to evaluate emotional anxiety and arousal showed significantly lower mean average deltas for the robotic surgeries than the open ones (Table 3, Table 4 and Table 5). More specifically, while for open surgeries the mean delta score of the STAI-Y questionnaire decreased—that is, anxiety decreased—for the robot-assisted surgery, the score increased, showing higher levels of anxiety. The SAM showed decreased arousal in both postsurgery conditions, but the postsurgery valence still leaned toward a negative activation.

The postsurgery evaluations of the performance showed that open surgery had higher scores for mental load, physical demand, frustration, perceived effort, and temporal demand than the robotic approach, but it scored better in perceived performance satisfaction.

The perceived physical comfort scores showed that robot-assisted surgery was significantly less taxing on the surgeons than the open approach.

## 4. Discussion

The autonomic nervous system is strictly engaged in the regulation of hemodynamic, metabolic, and respiratory responses to stress and is balanced between sympathetic and parasympathetic branches. The sympathetic ANS component is activated by stress, with subsequent chronotropic and inotropic effects on the heart; vasocontraction, resulting in the elevation of blood pressure; and increases in the respiratory rate, characterized by shallow breaths. On the other hand, the parasympathetic branch has the opposite effect [17]. The study of ANS responses to surgical stress has been the subject of several papers in which surgeons, in comparison to medical physicians, are subjected to harder work in uncomfortable and exhausting positions and stressful situations [11]. Heart rate, heart rate variability, blood pressure, O2 saturation, and the time of desaturation have been considered in various papers as markers of stress linked to surgical acts [11,12,13,14,15,16,17] that are regulated by noradrenergic pathways of sympathetic innervation on the cardiovascular and respiratory systems [19,20,21,22,23,24,25,26]. 

In this study, we evaluated the psychological and physiological responses to surgical stress of two experienced surgeons, both in open and in robotic surgery. 

In accordance with our cardiologic division, we used a wearable device (Healer R2) utilized for the remote multi-parametric monitoring of patients and for diagnostic exams in the hospital or at home. The use of this kind of device exploded during the COVID-19 era thanks to the possibility of controlling parameters remotely via the cloud.

It is widely known that the benchmarks of the ANS and of sympathetic activation are the heart and respiratory rate and blood pressure. However, we deliberately chose not to use blood pressure as a stress parameter because device mounting and the pumping by a Holter device every 10 min could in itself cause stress to the surgeon and compromise the surgical act, especially if prolonged and complex.

The first aspect emerging from our analysis was the important increase in heart rate during surgical activity for both surgeons in both surgical approaches compared to the baseline activity (Table 2). Dekker et al. [15] suggested that a high heart rate with a low HRV could precede cardiovascular disease manifestation and that a sympathetic predominance may be indicative of less favorable general health. They concluded that, in a population-based study of middle-aged men and women, a high heart rate was predictive of increased mortality rates. It is therefore self-evident that the stress associated with repeated surgical interventions can negatively affect the health of the surgeons themselves.

Ruitenburg et al. [11] asserted that surgeons, in comparison with other clinical physicians, perform fine repetitive movements 26 times longer and stand on their feet 130% longer. These prolonged repetitive movements associated with uncomfortable and exhausting postures make a surgeon’s work stressful and physically strenuous. 

Therefore, the robotic approach presents some important advantages compared to the traditional open or thoracoscopic approach. First of all, robotic instruments have various degrees of freedom, making it possible to replicate the movements of the traditional open technique. Secondly, it allows an important limitation of the natural tremor of a surgeon’s hands by converting movements into micro-movements. Thirdly, the surgeon sits comfortably at the console, controlling the instruments while viewing the operatory field in high-definition 3D. Last but not least, the console is ergonomic and adjustable, and the surgeon can adjust the heights of the optical viewer, arms, or pedal board. This represents an important advantage, especially in complex “high-fatigue” operations, where the surgeon assumes incongruous and physically strenuous postures for a long time.

From our analysis, the surgeons were subjected to greater stress in open surgery compared to robotic procedures. In particular, the heart and respiratory rates were higher during interventions performed via thoracotomy in both surgeons, as was the mean desaturation and the time in desaturation. The psychological evaluations confirmed the ANS responses. These aspects are linked for various reasons, which can be correlated both to the surgical approach itself and to the intervention.

It is obvious that the sitting posture (during robotic surgery) implies more comfort than the standing position (via thoracotomy); this could justify the greater and higher activation of the sympathetic system when the surgeon works via an open approach. Concerning the mean saturation and desaturation during surgery, the detection of these values could also be linked, above all, to the imperfect adherence of the sensors during the surgeon’s movements, especially in open surgery.

On the other hand, it is known that minimally invasive surgery is normally reserved for easier or less complex interventions. In our experience, the robotic approach is reserved for all initial I and II stage lung cancers (cN0 or cN1); the open approach is reserved for more complex cases (i.e., after neoadjuvant therapy, tumor dimensions >5 cm, chest wall involvement, or adjacency to mediastinal structures). It is quite intuitive that, in addition to the standing and the exhausting posture, the difficulty of the intervention also plays a decisive role in the stress linked to a surgical intervention. Neoadjuvant therapy for cN2 tumors represents an important factor conditioning the difficulty of the intervention; this aspect is linked to the post-therapy inflammatory reaction of pathological tissues and to the difficulty in easily dissecting them during the intervention. Despite the fact that we recently started a program for performing robotic interventions, even after neoadjuvant therapy (two lobectomies after chemotherapy and one after immunotherapy), almost half of the interventions via thoracotomy (55%) were performed after neoadjuvant therapy. It is interesting to note, therefore, that while the robot-assisted surgeries appear less taxing and demanding on the surgeons’ overall effort and ergonomics, performance satisfaction was rated higher for open-approach surgeries.

There were some limitations to our study. First of all, the interventions performed via thoracotomy were more complex (at least on paper) compared to those performed via the robotic approach. For this reason, in our analysis we included anatomical lung resections performed via thoracotomy, but we excluded more complex interventions (i.e., pleurectomy/decortication, extended pneumonectomies, and sleeve lobectomies). It is obvious that more complex cases can cause greater stress to the surgeon, and this could be a bias; at the same time, it is not ethically correct to propose a lung resection via thoracotomy in 2022 to a patient affected by early-stage lung cancer. Another limitation was the small number of the cases. In the future, we could corroborate the results of this study by comparing the different stress levels of a surgeon performing robotic and open surgeries for the same intervention (post-CT/immunotherapy lobectomy). As already reported, we recently started a robotic surgery project for post-CT/immunotherapy interventions.

## 5. Conclusions

We observed significantly increased levels of stress during surgical procedures for the surgeons, especially if they were performed using an open approach. The results correlated with a subjective assessment of stress (State-Trait Anxiety Inventory). For various reasons, some related to the method and others related to the intervention, the robotic approach led to less stimulation of the autonomic nervous system, producing less stress for the surgeons and ensuring greater comfort. 

## Figures and Tables

**Figure 1 cancers-15-01207-f001:**
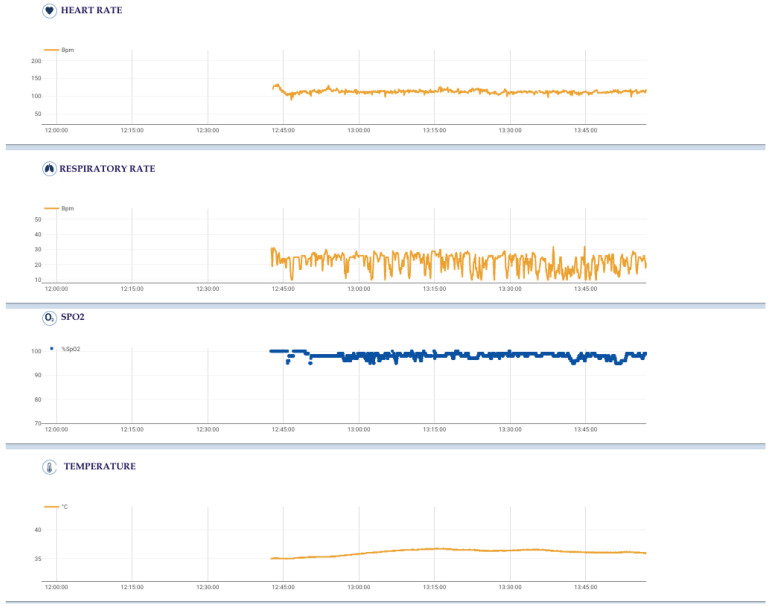
Physiological parameters of a single intervention (start time: 12:38, end time: 14:00), each with its own sampling frequency, acquired using the sensorized Healer R2 garment and elaborated using the Healer software.

**Figure 2 cancers-15-01207-f002:**
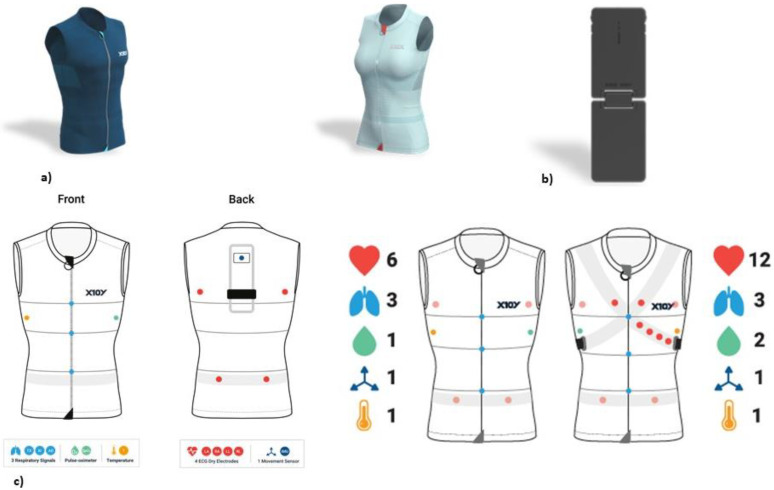
(**a**) Healer R2 wearable devices (male and female models); (**b**) Logger KoR 1, connecting to the back of the garment for storing the data and transferring them via 4G/5G/Wi-Fi; (**c**) embedded sensors to detect heart rate, respiratory rate, temperature, saturation, and desaturation.

**Table 1 cancers-15-01207-t001:** Pre- and postoperative outcomes concerning the types of interventions, the clinical histories of the patients, and complications.

	All Surgeons			Surgeon A			Surgeon B		
	Open	RATS	*p*-Value	Open	RATS	*p*-Value	Open	RATS	*p*-Value
	N (%)	N (%)		N (%)	N (%)		N (%)	N (%)	
**Patients**	40	40		20	20		20	20	
**Intervention**									
RLL	6	3		3	2		3	1	
LLL	4	6		2	4		2	2	
ML	1	2		0	1		1	1	
RUL	10	14		5	7		5	7	
LUL	10	15		6	6		4	9	
Right pneumonectomy	1	0		0	0		1	0	
Left pneumonectomy	1	0		0	0		1	0	
Bilobectomy	4	0		2	0		2	0	
Classic segmentectomy	3	0	0.04	2	0		1	0	
**Conversion**	0	1	-	0	1		0	0	
**Neoadjuvant CT**									
No	18	37		5	20		13	17	
Yes	22	3	<0.0001	15	0	<0.0001	7	3	0.27
**Comorbidities**									
Lung comorbidity	22	12	0.04	13	9	0.34	9	3	0.08
Cardiac comorbidity	23	21	0.82	12	13	1.00	11	8	0.53
Metabolic comorbidity	14	18	0.49	5	12	0.05	9	6	0.51
Other comorbidity	22	24	0.82	8	10	0.75	14	14	1.00
**Complications**									
No	29	35		15	15		14	20	
Yes	11	5	0.16	5	5	1.00	6	0	0.02
Air leak	5	2	0.43	3	2	1.00	2	0	0.49
Atrial fibrillation	4	2	0.68	0	2	0.15	4	0	0.11
Other complications	6	1	0.11	2	1	1.00	4	0	0.11
**Postoperative outcomes**	**Open**	**RATS**	***p*-value**	**Open**	**RATS**	***p*-value**	**Open**	**RATS**	***p*-value**
	mean ± SD	mean ± SD		mean ± SD	mean ± SD		mean ± SD	mean ± SD	
Hospitalization, days	8.3 ± 3.2	5.4 ± 2.3	**<0.0001**	7.2 ± 2.8	6.0 ± 3.1	0.20	9.4 ± 3.3	4.9 ± 1.0	**<0.0001**
Drainage, days	5.7 ± 2.5	4.2 ± 2.0	**0.003**	5.4 ± 3.2	4.7 ± 2.6	0.45	6.0 ± 1.5	3.6 ± 0.8	**<0.0001**
Intervention length, min	146 ± 50	124 ± 27	**0.02**	143 ± 38	130 ± 24	0.19	150 ± 61	119 ± 30	0.054

**Table 2 cancers-15-01207-t002:** Intraoperative respiratory and cardiovascular outcomes of the surgeons.

	All Surgeons			Surgeon A			Surgeon B		
	Open	RATS	*p*-Value	Open	RATS	*p*-Value	Open	RATS	*p*-Value
	Mean ± SD	Mean ± SD		Mean ± SD	Mean ± SD		Mean ± SD	Mean ± SD	
**Heart Rate at baseline**									
Mean	80.6 ± 3.0	80.5 ± 3.9	0.91	78.9 ± 2.9	77.6 ± 3.0	0.17	82.3 ± 2.1	83.4 ± 2.2	0.11
Min	75.5 ± 3.1	74.4 ± 3.1	0.15	74.4 ± 3.1	72.8 ± 2.8	0.11	76.6 ± 2.8	76.1 ± 2.5	0.56
Max	85.7 ± 4.3	86.6 ± 5.7	0.45	83.5 ± 3.8	82.4 ± 4.2	0.41	88.0 ± 3.5	90.7 ± 3.7	**0.02**
**Heart Rate during intervention**									
Mean	107.3 ± 9.1	95.6 ± 10.4	<0.0001	104.6 ± 5.8	88.1 ± 3.1	**<0.0001**	109.9 ± 11.0	103.0 ± 9.7	**0.04**
Mean difference	Diff = 11.7 ± 9.8			Diff = 16.5 ± 4.6			Diff = 6.9 ± 10.4		
Min	95.4 ± 9.8	84.3 ± 9.9	<0.0001	88.3 ± 6.4	78.3 ± 5.1	**<0.0001**	102.5 ± 7.0	90.3 ± 9.9	**<0.0001**
Max	123.7 ± 12.2	110.3 ± 10.2	<0.0001	118.1 ± 6.7	103.6 ± 5.1	**<0.0001**	129.3 ± 14.0	117.1 ± 9.5	**0.003**
**Respiratory frequency**									
Mean	18.0 ± 2.5	17.4 ± 4.4	0.42	15.7 ± 0.9	13.2 ± 1.6	**<0.0001**	20.3 ± 1.0	21.5 ± 1.2	**0.002**
Min	10.0 ± 0.2	9.8 ± 0.6	0.22	10.0 ± 0.0	9.8 ± 0.8	0.17	9.9 ± 0.3	9.9 ± 0.3	1.00
Max	27.8 ± 3.1	28.2 ± 4.6	0.65	25.1 ± 1.5	24.3 ± 3.0	0.29	30.6 ± 1.1	32.2 ± 1.3	**0.0002**
**Body temperature**									
Mean	36.2 ± 0.5	36.2 ± 0.3	0.69	36.2 ± 0.6	36.4 ± 0.3	0.23	36.3 ± 0.3	36.0 ± 0.2	**0.01**
Min	35.7 ± 0.3	35.8 ± 0.5	0.43	35.6 ± 0.4	35.8 ± 0.7	0.21	35.8 ± 0.3	35.8 ± 0.3	0.47
Max	36.9 ± 0.5	36.5 ± 0.4	0.0004	37.1 ± 0.4	36.7 ± 0.3	**0.001**	36.6 ± 0.5	36.3 ± 0.3	**0.008**
**Saturation**									
Mean	97.7 ± 0.9	97.7 ± 0.7	0.69	97.5 ± 1.2	98.1 ± 0.2	**0.06**	97.8 ± 0.4	97.4 ± 0.9	**0.07**
Min	90.8 ± 3.0	91.8 ± 2.5	0.12	89.4 ± 3.5	91.5 ± 3.1	**0.06**	92.2 ± 1.5	92.1 ± 1.9	0.85
Max	98.5 ± 0.8	98.6 ± 0.6	0.53	98.4 ± 1.0	98.7 ± 0.5	0.34	98.6 ± 0.5	98.6 ± 0.7	0.79
**Desaturation**									
Duration	214 ± 153	227 ± 326	0.83	106 ± 69.7	85.7 ± 61.0	0.34	323 ± 135	368 ± 415	0.65
Mean value	3.8 ± 0.4	3.7 ± 0.6	0.53	3.7 ± 0.5	3.4 ± 0.7	**0.08**	3.8 ± 0.4	4.0 ± 0.0	**0.04**

**Table 3 cancers-15-01207-t003:** Means and standard deviations (in parentheses) for SAM dimensions and STAI-Y delta scores.

	SAM Arousal	SAM Valence	STAI-Y 1
Open Approach	−1.00 (0.94)	1.30 (1.06)	−1.30 (5.85)
Robot-assisted approach	−0.33 (0.62)	0.27 (0.59)	3.67 (3.24)
Total	−0.60 (0.82)	0.68 (0.95)	1.68 (5.01)

**Table 4 cancers-15-01207-t004:** Means and standard deviations (in parentheses) for delta scores.

	Neck	Right Shoulder	Left Shoulder	Upper Back	Lower Back	Left Hand/Wrist	Right Hand/Wrist
Open Approach	2.50 (2.12)	2.30 (2.06)	2.50 (2.42)	3.40 (2.72)	3.50 (2.88)	0.50 (1.27)	0.10 (0.32)
Robot-assisted approach	1.00 (1.25)	0.47 (0.99)	0.33 (0.90)	1.07 (1.58)	1.07 (1.71)	0.47 (0.83)	0.47 (0.83)
Total	1.60 (1.78)	1.20 (1.73)	1.20 (1.96)	2.00 (2.36)	2.04 (2.51)	0.48 (1.00)	0.32 (0.69)

**Table 5 cancers-15-01207-t005:** Means and standard deviations (in parentheses) for the dimensions of the NASA TLI.

	Mental Demand	Physical Demand	Frustration	Temporal Demand	Effort	Performance
Open Approach	15.80 (4.47)	12.10 (5.51)	7.20 (4.29)	11.00 (4.76)	12.10 (5.34)	13.60 (4.06)
Robot-assisted approach	9.67 (5.00)	4.80 (3.30)	2.93 (1.44)	7.00 (3.64)	7.07 (3.37)	9.60 (4.45)
Total	12.12 (5.61)	7.72 (5.44)	4.64 (3.56)	8.60 (4.50)	9.08 (4.86)	11.20 (4.66)

## Data Availability

The original contributions presented in this study are included in the article.

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
