# Peer review of "How Much Stress Does a Surgeon Endure? The Effects of the Robotic Approach on the Autonomic Nervous System of a Surgeon in the Modern Era of Thoracic Surgery"

_cancers, 2023, doi:10.3390/cancers15041207_

Round 1

Reviewer 1 Report

First of all, this is a very relevant topic, that is relatively neglected but is nevertheless gaining more interest.

The authors could have done a better job presenting the subject. The authors outline a number of different concepts in the introduction. “Emotional arousal stress”, “general comfort”, “psychological response to stress”, “cardiovascular stress”, “physically strenuous”, “uncomfortable and exhausting positions”, “physiologic impact of surgical stress”. However, none of these is clearly defined, the relationship between them is not discussed, and it is not explicitly stated which of these is the objective of the paper. The authors could also have elaborated on the relevance of looking at surgeon stress. In the discussion, they state that it is “self evident” that increased physiologic arousal during surgery can negatively affect the health of surgeons. I would argue that it is hardly self evident; rather, it is the responsibility of the authors to make the argument why this would be so, by basing their argument on an interpretation of their results in light of existing evidence.

There are indeed existing data on a variety of surgeons’ physiologic parameters during operative interventions. Instrument companies use such parameters routinely in the development of new surgical technologies. In addition, these physiologic parameters vary not just by surgical approach, but they are also responsive to specific steps of an operation, some of which may be more “stressful” than others. Companies often focus their instrument development specifically to address these stressful steps. So a more thorough literature review could have helped inform this study and maybe helped explain some of the results and observed differences between groups.

Since open and robotic surgery are fundamentally different, it would have been very important to define what is being looked at. For example, it is not clear whether the observed physiologic changes reflect “psychological stress” or rather reflect bad ergonomics and increased physical demands. So it all comes down to a question of validity: unless what we are looking at is clearly defined, the results will be difficult to interpret.

The surgeries being performed open and robotic in this study are fundamentally different. We therefore end up with experimental and control groups that are not comparable. This makes it impossible to attribute the observed results to the approach itself. No matter how the authors improve the paper, I don’t see how this limitation can be overcome, since it is an inherent to the study design.

It is not clear to me why stress parameters were increased while subjective anxiety was decreased in open vs robotic surgery. The authors did not explain this discrepancy. Also, it is not clear to me why oxygen saturation would decrease. Even in strenuous and sustained physical exertion, saturation does not decrease; a decrease in saturation is usually reflective of poor cardiopulmonary reserve. These are just two examples of the kind of important information that is lacking in this paper.

I find it a bit surprising that no ethics approval was required for a prospective study. Wearing the vest has implications; it can certainly increase the level of psychologic stress, and can possibly result in some limitation of movement for the surgeon. Both of these issues may in theory impact the quality of the surgery, regardless of the apparently low risk.

In the conclusion, the authors state that: “For various reasons, some related to the method, others related to the intervention, the robotic approach leads to less stimulation of the autonomic nervous system, producing less stress for the surgeons and ensuring greater comfort.” Such a statement is not informative. It is the responsibility of the authors to outline those “various reasons”, to explain their effect, their relevance, how they bear on the results, and what role those variables should have in future studies.

In summary, I think that the subject matter is relevant but the study as designed and reported has several major shortcomings. The fact that the experimental and control groups are fundamentally different may be an insurmountable limitation that unfortunately undermines the possibility of any useful take away.

Author Response

REVIEWER 1

First of all, this is a very relevant topic, that is relatively neglected but is nevertheless gaining more interest.

Question n.1/ answer n.1

The authors could have done a better job presenting the subject. The authors outline a number of different concepts in the introduction. “Emotional arousal stress”, “general comfort”, “psychological response to stress”, “cardiovascular stress”, “physically strenuous”, “uncomfortable and exhausting positions”, “physiologic impact of surgical stress”. However, none of these is clearly defined, the relationship between them is not discussed, and it is not explicitly stated which of these is the objective of the paper. The authors could also have elaborated on the relevance of looking at surgeon stress. In the discussion, they state that it is “self evident” that increased physiologic arousal during surgery can negatively affect the health of surgeons. I would argue that it is hardly self evident; rather, it is the responsibility of the authors to make the argument why this would be so, by basing their argument on an interpretation of their results in light of existing evidence.

Thanks for the remarks.

We tried to evaluate the pre-operative stress of a surgeon, evaluating both the psychological and physical aspects (the first one evaluated by pre and post-operative questionnaires; the second ones evaluating the variations related to the activation of the sympathetic system). In the discussion section, we largely argument the association between stress, surgery and activation of sympathetic system  “……the sitting posture  implies more comfort than the standing position……this could justify the greater and higher activation of the sympathetic system when the surgeon works via open approach……..minimally invasive surgery is normally reserved for easier or less complex interventions…… the difficulty of the intervention also plays a decisive role in the stress linked to surgical intervention”.

On the other hand, the concept of “arousal stress” or the psychological response to the stress of each surgeon, is difficult to make objective, except with pre-operative or post-operative tests. The real psychological response to stress during the surgical act, unfortunately is impossible to evaluate (we would need a specialist who evaluates us during the surgical act and this could greatly affect the surgical act itself). The sentence you are referring to is: “It is therefore self-evident that the stress associated with repeated surgical interventions can negatively affect the health of the surgeons themselves”. I avoided discussing about the reasons why a job, characterized by constant and high repeated stress could negatively affect the health. I think that it’s a common notion that a physical and mental high demanding work, can determine negative effects on physical and psychological health of the worker. This aspect is very stressed in other papers (Jones KI, Amawi F, Bhalla A, Peacock O, Williams JP, Lund JN. Assessing surgeon stress when operating using heart rate variability and the State Trait Anxiety Inventory: will surgery be the death of us? Colorectal Dis. 2015 Apr;17(4):335-41. doi: 10.1111/codi.12844. PMID: 25406932.)

Question n.2/ answer n.2

There are indeed existing data on a variety of surgeons’ physiologic parameters during operative interventions. Instrument companies use such parameters routinely in the development of new surgical technologies. In addition, these physiologic parameters vary not just by surgical approach, but they are also responsive to specific steps of an operation, some of which may be more “stressful” than others. Companies often focus their instrument development specifically to address these stressful steps. So a more thorough literature review could have helped inform this study and maybe helped explain some of the results and observed differences between groups.

Our idea was to evaluate how the robotic approach could positively influence surgical performance, taking into account the surgeon and his abilities. Thus, we took into consideration the few works analysing the intraoperative cardiovascular parameters of the surgeon. Unfortunately, although many companies study the ergonomics and comfort of their instruments (as rightly suggested by the reviewer), the literature about is scarse and poor of works; the only data found in the recent years were those reported in the manuscript. Precisely for this reason, despite the limitation linked to the difference in interventions, we have decided to follow this path and we will expand the study also to post CT/immunotherapy lung resections via robotic approach.

Question n.3/ answer n.3

Since open and robotic surgery are fundamentally different, it would have been very important to define what is being looked at. For example, it is not clear whether the observed physiologic changes reflect “psychological stress” or rather reflect bad ergonomics and increased physical demands. So it all comes down to a question of validity: unless what we are looking at is clearly defined, the results will be difficult to interpret.

Thanks for the observation. The physiologic changes are direct consequences both of psychological stress and bad ergonomics and increased physical demands. We reported these aspects in the discussion section

  • “These aspects are linked to various reasons, which can be correlated both to the surgical approach itself and to the intervention……”
  • “It is obvious that the sitting posture (during robotic surgery) implies more comfort than the standing position (via thoracotomy); this could justify the greater and higher activation of the sympathetic system when the surgeon works via open approach”
  • “On the other hand, it is known that minimally invasive surgery is normally reserved for easier or less complex interventions”
  • “It is quite intuitive that, in addition to the standing and exhausting posture, the difficulty of the intervention also plays a decisive role in the stress linked to surgical intervention”

Question n.4/ answer n.4

The surgeries being performed open and robotic in this study are fundamentally different. We therefore end up with experimental and control groups that are not comparable. This makes it impossible to attribute the observed results to the approach itself. No matter how the authors improve the paper, I don’t see how this limitation can be overcome, since it is an inherent to the study design.

Thanks for the observation. I agree with you and we add the limitations in the discussion section.

There are some limitations in our study. First of all, the interventions performed via thoracotomy, were more complex (at least on paper), compared to those via robotic approach. For this reason, in our analysis we included anatomical lung resection performed via thoracotomy, but we excluded more complex interventions (i.e. plurectomy/decortication, extended pneumonectomies, sleeve lobectomies). It’s obvious that more complex cases can cause greater stress to the surgeon and this could be a bias; at the same time, it is not ethically correct proposing a lung resection via thoracotomy in 2022 to a patient affected by early stage lung cancer. Another limitation is the small number of the cases. In the future, we could corroborate the results of this study, comparing the different stress levels of a surgeon, performing robotic and open surgery for the same intervention (post-CT/immunotehrapy lobectomy). Indeed, we have recently started a robotic surgery project for post CT/immunotherapy interventions.

Question n.5/ answer n.5

It is not clear to me why stress parameters were increased while subjective anxiety was decreased in open vs robotic surgery. The authors did not explain this discrepancy. Also, it is not clear to me why oxygen saturation would decrease. Even in strenuous and sustained physical exertion, saturation does not decrease; a decrease in saturation is usually reflective of poor cardiopulmonary reserve. These are just two examples of the kind of important information that is lacking in this paper.

Thanks for the remark. The questionnaires for the evaluation of stress, were drawn up by individual surgeons before surgery (therefore based on the idea of what the intervention would be). Unfortunately, the real intraoperative psychological stress  cannot be evaluated (we cannot evaluate, during a surgery, the psychological reaction of the individual surgeon). This is to say that, sometimes, the idea of something that causes anxiety, does not really determine that stress when you face it.

Concerning desaturation, we evaluate maximum and minimum saturation and the time of desaturation and in the analysis we only evaluated mean desaturation (maximum 3 or 4 percentage points. In addition, sometimes, the value of saturation could be linked also and above all to the imperfect adherence of the sensors during the surgeon's movements, especially in open surgery. We add this aspect into discussion section.

Question n.6/ answer n.6

I find it a bit surprising that no ethics approval was required for a prospective study. Wearing the vest has implications; it can certainly increase the level of psychologic stress, and can possibly result in some limitation of movement for the surgeon. Both of these issues may in theory impact the quality of the surgery, regardless of the apparently low risk.

Thanks for the observation. As already reported in the manuscript, we consulted our Ethical Committee before starting the project. The device (Healer LIFE Homepage - L.I.F.E. (x10x.com) ) is already used and approved in real life (both for patients and even for the athletes) without  any impairment of normal daily activities or performance, both in basal conditions and under stress. In fact, there was no discomfort to wear the device during the interventions (who writes at this time is one of the two surgeons in the study). On the other hand, precisely to avoid influencing the surgical act, we did not use blood pressure as a stress parameter (as already widely described in the material & methods section: “We did not consider blood pressure as a parameter for different reasons; first of all, mounting and pumping by a Holter device every 10 minutes could in itself cause stress to the surgeon. The second aspect is the possible damage linked to the pumping to the artery; lastly, continuous and intermittent pumping could compromise the surgical act, especially if prolonged and complex”).

Question n.7/ answer n.7

In the conclusion, the authors state that: “For various reasons, some related to the method, others related to the intervention, the robotic approach leads to less stimulation of the autonomic nervous system, producing less stress for the surgeons and ensuring greater comfort.” Such a statement is not informative. It is the responsibility of the authors to outline those “various reasons”, to explain their effect, their relevance, how they bear on the results, and what role those variables should have in future studies.

Thanks for the observation, The reviewer quotes our final sentence in the conclusions section, where we must be concise and summarize in one sentence the concept and results of the work. In the discussion these various reasons are largely and widely discussed:

  • “These aspects are linked to various reasons, which can be correlated both to the surgical approach itself and to the intervention……”
  • “It is obvious that the sitting posture (during robotic surgery) implies more comfort than the standing position (via thoracotomy); this could justify the greater and higher activation of the sympathetic system when the surgeon works via open approach”
  • “On the other hand, it is known that minimally invasive surgery is normally reserved for easier or less complex interventions”
  • “It is quite intuitive that, in addition to the standing and exhausting posture, the difficulty of the intervention also plays a decisive role in the stress linked to surgical intervention”

In summary, I think that the subject matter is relevant but the study as designed and reported has several major shortcomings. The fact that the experimental and control groups are fundamentally different may be an insurmountable limitation that unfortunately undermines the possibility of any useful take away.

Thanks for your comments. They have been precious and useful to improve our project. We hope that with our adjustments and explanations, our work is now worthy of publication.

Reviewer 2 Report

The topic of this work is interesting, exploring the physical and psychological impact of thoracic surgery on the practitioner.

In my opinion, this paper is certainly innovative since there is no trace of similar studies in the literature for this surgical branch.

The abstract is adequate, summarizes the essential aspects of the manuscript and is easily readable.

Perhaps, I suggest adding to the conclusions that, although open surgery has a more obvious impact, it gives the surgeon greater perceived performance satisfaction.

The introduction provides the right background for understanding the study; the works quoted are adequate.

The materials and methods present the setting of the study in a clear and reproducible way.

The results are consistent with the initial hypothesis of the study and the number of tables is adequate.

The discussion is of the right length and the studies in the literature have been correctly cited.

Overall, the study is worthy of publication.

I have only two questions for the authors:

- since open surgery has been reserved for complex cases and which, therefore, can cause greater stress to the surgeon, do you think this could be a bias? I obviously realize that it is not ethically correct to subject a patient to open surgery if susceptible to the RATS technique (i.e. an early stage)

- a part of the study is based on the subjective component of the surgeon, quantified through questionnaires. In your opinion, can the inter-individual variability of the psychological set-up influence the results obtained? Expanding the number of surgeons evaluated, perhaps with a future multi-center study, what kind of results would you expect?

Author Response

REVIEWER 2

The topic of this work is interesting, exploring the physical and psychological impact of thoracic surgery on the practitioner.

In my opinion, this paper is certainly innovative since there is no trace of similar studies in the literature for this surgical branch.

The abstract is adequate, summarizes the essential aspects of the manuscript and is easily readable.

Perhaps, I suggest adding to the conclusions that, although open surgery has a more obvious impact, it gives the surgeon greater perceived performance satisfaction.

The introduction provides the right background for understanding the study; the works quoted are adequate.

The materials and methods present the setting of the study in a clear and reproducible way.

The results are consistent with the initial hypothesis of the study and the number of tables is adequate.

The discussion is of the right length and the studies in the literature have been correctly cited.

Overall, the study is worthy of publication.

Thanks to reviewers for the analysis of the manuscript.

I have only two questions for the authors:

- since open surgery has been reserved for complex cases and which, therefore, can cause greater stress to the surgeon, do you think this could be a bias? I obviously realize that it is not ethically correct to subject a patient to open surgery if susceptible to the RATS technique (i.e. an early stage)

I thanks the reviewer. Open surgery deserved more critical and difficult intervention, comparing to the robotic approach. In our analysis we included lung resection performed via thoracotomy, but we excluded interventions more complex, such as decortication, intrapericardial pneumonectomies, sleeve lobectomies, etc. I agree with the reviewers; more complex cases can cause greater stress to the surgeon and this could be a bias; at the same time, it is not ethically correct to propose an open surgery to an ealry stage in 2022. In the future, we could compare the different stress levels of a surgeon, performing robotic and open surgery for the same intervention: post-CT lobectomy. We add some sentences into discussion section for clarifying these aspects.

- a part of the study is based on the subjective component of the surgeon, quantified through questionnaires. In your opinion, can the inter-individual variability of the psychological set-up influence the results obtained? Expanding the number of surgeons evaluated, perhaps with a future multi-center study, what kind of results would you expect?

I totally agree with the reviewer. Even if, buying different devices could be expensive, this idea is now in progress for the next research project.

Reviewer 3 Report

The authors reported an interesting paper on the effect of surgery on ANS of surgeons and they compared the effects of robotic and open surgery. The study is interesting and well written. I really enjoyed to read it. I have minor comments:

- Do some of the surgeon take some drugs? It should be reported in the methods section

- Will the results of this study change something in your current standard? Please comment on this.

- Did you evaluate a possible correlation between surgeons' performances and postoperative outcomes (e.g. postoperative complications)?

Author Response

REVIEWER 3

The authors reported an interesting paper on the effect of surgery on ANS of surgeons and they compared the effects of robotic and open surgery. The study is interesting and well written. I really enjoyed to read it. I have minor comments:

- Do some of the surgeon take some drugs? It should be reported in the methods section

Thanks for the remark. No surgeon was taking medication at the time of the study. We add this sentence in the material and methods section.

- Will the results of this study change something in your current standard? Please comment on this.

The results of this study corroborate our idea that robotic approach is comfortable, sure and safe, and, also following these results, we started a project of robotic surgery post CT/immunotherapy, previously approached by open surgery.

- Did you evaluate a possible correlation between surgeons' performances and postoperative outcomes (e.g. postoperative complications)?

Thanks for the observation. We took into account this aspect but we after gave up. In my opinion, the only complication that could be related to the performance of the surgeon is the persistence of air leaks (not atrial fibrillation or other medical complications); however, even in this case, the causes could be (COPD, fragile parenchyma, staplers disfunction...). Precisely for this reason, we have not evaluated this correlation.

Round 2

Reviewer 1 Report

I appreciate the authors’ work in revising their paper. However, the changes are limited and do not address my concerns with the initial submission in any fundamental way. In my opinion this paper does not meet the standard for publication.